# Exploiting a cognitive bias promotes cooperation in social dilemma experiments

Zhen Wang [1], Marko Jusup [2,3], Lei Shi[4], Joung-Hun Lee[5], Yoh Iwasa [6,7] & Stefano Boccaletti[8,9]

The decoy effect is a cognitive bias documented in behavioural economics by which the presence of a third, (partly) inferior choice causes a significant shift in people's preference for other items. Here, we performed an experiment with human volunteers who played a variant of the repeated prisoner's dilemma game in which the standard options of "cooperate" and "defect" are supplemented with a new, decoy option, "reward". We show that although volunteers rarely chose the decoy option, its availability sparks a significant increase in overall cooperativeness and improves the likelihood of success for cooperative individuals in this game. The presence of the decoy increased willingness of volunteers to cooperate in the first step of each game, leading to subsequent propagation of such willingness by (noisy) tit-for-tat. Our study thus points to decoys as a means to elicit voluntary prosocial action across a spectrum of collective endeavours.

[1] School of Mechanical Engineering and Center for OPTical IMagery Analysis and Learning (OPTIMAL), Northwestern Polytechnical University, Xi'an 710072, China. [2] Center of Mathematics for Social Creativity, Hokkaido University, Sapporo 060-0812, Japan. [3] Institute of Innovative Research, Tokyo Institute of Technology, Tokyo 152-8552, Japan. [4] Statistics and Mathematics College, Yunnan University of Finance and Economics, Kunming 650221, China. [5] Institute of Decision Science for a Sustainable Soceity, Kyushu University, Fukuoka 819-0395, Japan. [6] Faculty of Science, Kyushu University, Fukuoka 819-0395, Japan. [7] Department of Bioscience, School of Science and Technology, Kwansei-Gakuin University, Sanda 669-1337, Japan. [8] Institute for Complex Systems of the CNR, Florence 50019, Italy. [9] Unmanned Systems Research Institute, Northwestern Polytechnical University, Xi'an 710072, China. These authors contributed equally: Zhen Wang, Marko Jusup. Correspondence and requests for materials should be addressed to Z.W. (email: w-zhen@nwpu.edu.cn) or to M.J. (email: mjusup@gmail.com) or to L.S. (email: lshi@ynufe.edu.cn)

Neoclassical economics has ascribed human actions to a relentless rational drive to maximise the (expected) utility[1–5], even as the economic models struggled to account for the full range of displayed behaviours[6–8]. The apparent discrepancy between theory and reality ultimately gave birth to the field of behavioural economics, which has since amassed indisputable evidence to the effect that various cognitive biases stand in the way of the completely rational behaviour of economic agents[9–12]. Our focus is on the decoy effect, also known as the asymmetric dominance effect[13] or the attraction effect[14], which appears in multiple-choice situations in which a particular choice called "decoy" shares some defining characteristics with another choice called "target", but is inferior to the target in one defining characteristic[15]. Because we apply the concept of decoy in the novel context of evolutionary game theory, it is crucial to establish which defining (shared and inferior) characteristics are relevant to this context, as will be done in a moment. We emphasise for now that while decoys should be inconsequential to decision-making, they have been found to increase the attractiveness of the target, even in real-world political elections[16]. The effect's strength, however, seems to diminish in the presence of more meaningful stimulus descriptions[17]. Motivated by this finding, multiple recent studies questioned the limits of validity of the decoy effect[18–21].

Apart from behavioural economics, evolutionary game theory is another research field in which considerable attention has been devoted to the study of human behaviour, offering, for instance, an updated view on cognitive biases[22–24]. The work on the evolution of human cooperation in particular boasts a rich mathematical modelling legacy[25,26] complemented by a more recent track of social dilemma experimentation[27–32], as well as somewhat rarer attempts to reconcile theoretical and empirical perspectives[33–35]. While the basic concepts are shared with economics (cf., fitness vs. utility), evolutionary game theory introduces an extra explanatory dimension behind fitness maximisation via the process of selection. Specifically, selection describes temporal evolution towards maximum fitness during which human reasoning faculties eliminate suboptimal behaviours in a trial-and-error manner. It is largely unknown, however, if the process of selection can be manipulated by means of cognitive biases.

We set up an experimental attempt within the bounds of evolutionary game theory designed to observe the decoy effect at work. Specifically, we organised a repeated Prisoner's Dilemma (rPD) game which, based on both theory and previous experience[32], should cause defection to considerably exceed cooperation. We then enriched this game with a third, decoy option to "throw off" players and get them to cooperate with one another early in the game. Finally, we asked whether such an initial burst of cooperation can be stabilised or whether players recognise that from a purely rational perspective the decoy option is irrelevant, leading to a gradual replacement of cooperation with mutual defection.

We recruited 388 undergraduate volunteers (mean age = 19.8, 67.8% women) to engage in the rPD game consisting of random pairwise encounters. Each encounter lasted, on average, four rounds because after any given round, a computer system could signal the end of the encounter with 25% probability. We allowed experimental sessions to run for about 75 min, leading to approximately 20 encounters per session. The purpose of these sessions was to contrast a control treatment, wherein the only actions available were cooperation (C) and defection (D), with a decoy treatment, wherein an additional action to "reward" one another (R) was also available. We presented the basic rules of the rPD game to volunteers in a neutrally framed manner using the

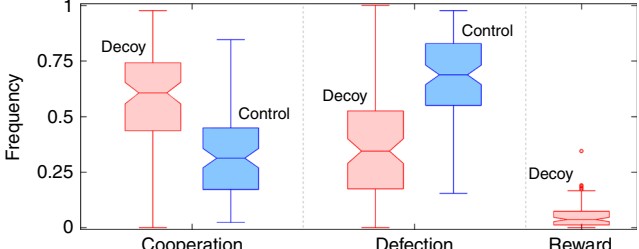

**Fig. 1** Reward ignites cooperativeness. Median frequency of cooperation equal to 60.5% in the decoy treatment is significantly higher (one-tailed Mann–Whitney U test; z-score 5.6713; p value $< 10^{-8}$) than the median frequency of cooperation equal to 31.4% in the control treatment. Conversely, the median frequency of defection equal to 34.5% in the decoy treatment is significantly lower (one-tailed Mann–Whitney U test; z-score $-6.3270$; p value $< 10^{-9}$) than the median frequency of defection equal to 68.6% in the control treatment. These results suggest that reward as the decoy option plays an instrumental role in eliciting cooperative behaviour. Despite this instrumental role, opponents reward one another with the median frequency of only 3.70%. Box-and-whisker plots with notches characterise the empirical distribution of action frequencies, obtained by counting, for each volunteer, the number of cooperative, defecting, or rewarding actions taken and then dividing these counts by the total number of rounds played. Box height determines the interquartile range, while the horizontal line inside the box represents the median. Notches indicate the 95% confidence intervals for the median. Whiskers span would encompass 99.3% of data if the data were normally distributed. Points outside of this span are drawn as outliers

following unilateral and bilateral payoff matrices:

$$
\begin{array}{cc}
\text{Own} & \text{Foe's}
\end{array}
\qquad
\begin{array}{ccc}
C & D & R
\end{array}
$$

$$
\begin{array}{c}
C \\ D \\ R
\end{array}
\begin{pmatrix}
-1 & 2 \\
1 & -1 \\
-2 & 3
\end{pmatrix}
\Rightarrow
\begin{array}{c}
C \\ D \\ R
\end{array}
\begin{pmatrix}
1 & -2 & 2 \\
3 & 0 & 4 \\
0 & -3 & 1
\end{pmatrix}
\qquad (1)
$$

In particular, action 1 (i.e., C) meant giving up one unit for the opponent to receive two units. Action 2 (i.e., D) meant earning one unit at the opponent's expense of one unit. Finally, action 3 (i.e., R) was qualitatively the same as action 1 with the distinction that one would give up two units for the opponent to receive three units in return. Further details on the experimental methods are given in the Methods section and the Supplementary Methods (see also Supplementary Figs. 1–3 and Supplementary Table 1). A picture emerging from this experimental setup is that reward is effective in promoting cooperation, but this effectiveness is a consequence of a cognitive bias known as the decoy effect. We therefore conclude that decoys possess an untapped potential to elicit voluntary prosocial action.

## Results

**Reward ignites cooperativeness.** The presence of reward R in the rPD game ignites cooperativeness (Fig. 1). Compared to the control treatment, the frequency of cooperation C (defection D) in the decoy treatment is significantly higher (lower). The frequency of C is furthermore stable in time (Fig. 2). An immediate implication is that reward plays an instrumental role in promoting cooperation, yet opponents seldom use the opportunity to reward one another. Because C and R share the property of being cooperative actions, this kind of behaviour is consistent with the decoy effect as described by behavioural economists[12,15], provided that R is also an inferior alternative to C.

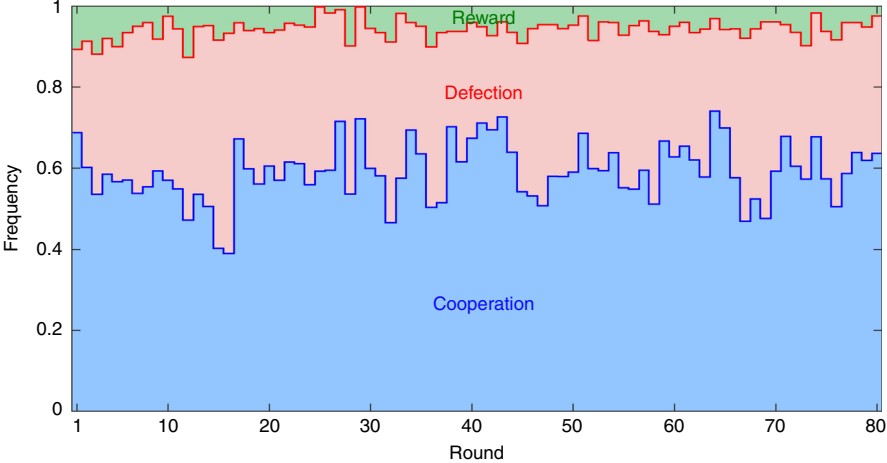

**Fig. 2** Initial burst of cooperation caused by the decoy option is stable in time. Shown are the frequencies of all three actions as they evolve through time in the decoy treatment. These frequencies were obtained by counting how many volunteers chose a particular action divided by the total number of volunteers playing. Treating the frequency of cooperation C as a time series, we executed the augmented Dickey–Fuller test to examine the statistical stationarity of this series. Denoting the frequency of C at time t with $C_t$, the augmented Dickey–Fuller test consisted of two steps. First, we fitted equation $C_t - C_{t-1} = \beta_1 + \beta_2 t + \beta_3 C_{t-1} + \beta_4 (C_{t-1} - C_{t-2}) + \epsilon_t$ to the displayed data, where $\beta_i$, $i = 1, 2, 3, 4$ are the regression coefficients, and $\epsilon_t$ is a normally distributed error term with a zero mean and an unknown variance. Second, we examined which regression coefficients were significantly different from zero. The results show that $\beta_1 = 0.4258 > 0$ (t test; t-statistic 5.420; $p < 10^{-6}$), $\beta_2 = 0.0005 \approx 0$ (t test; t-statistic 1.536; p value 0.129), $\beta_3 = -0.7586 < 0$ (t test; t-statistic −5.583; p value $< 10^{-6}$), and $\beta_4 = 0.0768 \approx 0$ (t test; t-statistic 0.674; p value 0.502). The negative value of $\beta_3$ reflects a self-correcting nature (i.e., stationarity) of the examined time series, with the cooperation frequency fluctuating around $\beta_1/(1-(1+\beta_3)) = 0.5613$. Additionally, the time series shows no significant trend ($\beta_2 \approx 0$) nor any autoregressive structure ($\beta_4 \approx 0$)

How do we generally determine which action choice is worse and which is better? If our interest was solely in single-shot games, then Eq. (1) would indicate that—in a game-theoretic sense—defection D dominates over cooperation C, and both D and C dominate over reward R. This is seen from the bilateral payoff matrix in which payoffs associated with D (C) are higher than the corresponding payoffs associated with C (R). Unfortunately, game repetitions complicate matters, forcing us to consider (i) the nature of the social dilemma and (ii) the effect of repetitions. The nature of the dilemma is distilled in the concept of dilemma strength[25,36]. For example, by taking a difference between payoffs obtained for successful defection and mutual cooperation, and then normalising by the difference in payoffs obtained for mutual cooperation and mutual defection, we quantify how "lucrative" defection is relative to cooperation. Accordingly, the stronger the dilemma, the easier it gets for defectors to exploit cooperators (Supplementary Note 1; Supplementary Fig. 4). The effect of repetitions, by contrast, is to potentially change the nature of the dilemma[37], at least in a probabilistic sense (Supplementary Note 2; Supplementary Fig. 5). This means that instead of having defection D as the only evolutionarily stable strategy (ESS), cooperation C may also become an ESS. For cooperators to prevail, their initial fraction $x_C^*$ must be sufficiently high to provide protection against excessive exploitation by defectors. This initial fraction is given by the dilemma strength parameter, DS, such that $x_C^* = \frac{q\mathrm{DS}}{1-q}$, where q is the game termination probability. The higher the value of DS, the higher the initial fraction of cooperators must be for them to prevail.

To apply the concept of dilemma strength for our purpose, we recognise from Eq. (1) that R is another form of cooperation, more costly than C, but also more beneficial for the opponent, and we look at which of the two actions better protects cooperators from excessive exploitation by defectors. Rationality dictates ignoring the form of cooperation that offers less protection, meaning that rational opponents should end up playing a 2 × 2 game consisting of either the (C, D) pair or the (D, R) pair. We find from Eq. (1) that for these pairs DS = 2 and DS = 3, respectively. A higher value of DS for pair (D, R) points to a less favourable dilemma and forces a conclusion that R is inferior to C. We thus learn that C and R share the defining characteristic of being cooperative actions, but R is inferior to C in another defining characteristic, i.e., resilience to exploitation by defectors. In the context of evolutionary game theory, therefore, R as defined in Eq. (1) is a valid decoy for C.

**Reward's effectiveness and a cognitive bias**. Searching for mechanisms that explain improved cooperativeness in the decoy treatment, we find that reward is an effective cooperation promoter even before it can be used. Compared to the control treatment, the odds of an encounter starting with cooperation C (defection D) are significantly improved (suppressed) (Fig. 3a). Past the first round, response to C (D) in the previous round is overwhelmingly C (D) irrespective of the treatment (Fig. 3b, c). Reward R is mostly met with C or an occasional R (Fig. 3d). These results suggest that volunteers in our experiment play what can be characterised as noisy tit-for-tat (TFT) (Fig. 4).

In evolutionary game theory, cooperativeness prevails if it leads to success in terms of fitness or payoff. We find that in the control treatment, the average payoff per-round correlates negatively (positively) with cooperation C (defection D). This situation improves significantly in the decoy treatment, but the improvement is insufficient to make the average payoff per-round positively (negatively) correlated with C (D) (Fig. 5a, b). Our failure to clearly show that cooperativeness leads to success in the decoy treatment is puzzling and raises questions about the mechanisms underlying selection, which we address below. Interestingly, the average payoff per-round shows no correlation with R either (Fig. 5c). This should be compared with punishment in previous experiments[29,30,32], wherein the performance of frequent punishers was dismal. Furthermore, the cooperation-promoting effect of punishment in these experiments was unreliable[30,32] (but also see Ref. 38).

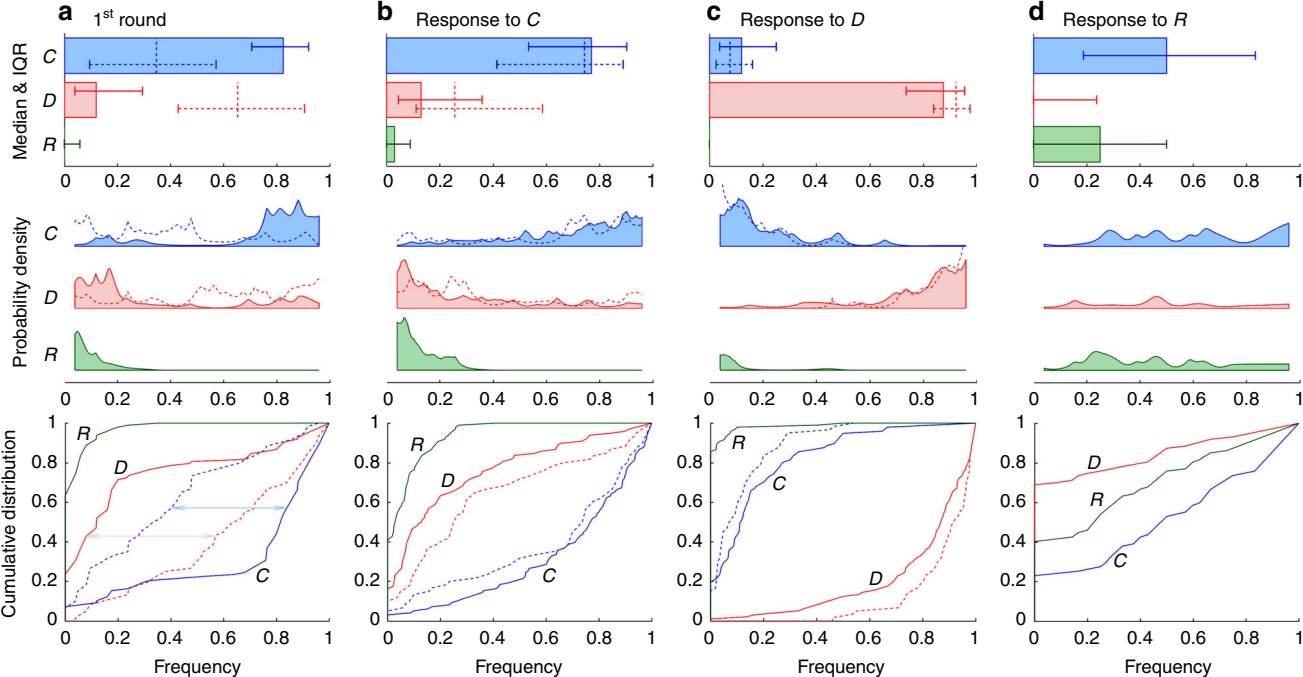

**Fig. 3** As a cooperation promoter, reward is effective even before it can be used. **a** Median frequency of cooperation C in the first round of the decoy treatment (bars) is considerably higher than in the control treatment (dashed lines): 82.4% vs. 34.7%, respectively. The opposite is true of defection D: 12.0% vs. 65.3%, respectively. Volunteers in the decoy treatment also appear to be more decisive than in the control treatment as hinted by a much narrower interquartile range: e.g., for C: 21.4% vs. 47.6%, respectively. Action frequencies were calculated as in Fig. 1. Probability densities show the frequency distributions, while cumulative densities reveal how distant these distributions are when decoy is compared to control (two-sample Kolmogorov–Smirnov test for C; K–S statistic 0.5772; p value < $10^{-10}$). **b–d** Past the first round, response to C in the previous round is overwhelmingly C irrespective of the treatment (two-sample Kolmogorov–Smirnov test for C; K–S statistic 0.1184; p value 0.6467), response to D in the previous round is overwhelmingly D irrespective of the treatment (two-sample Kolmogorov–Smirnov test for D; K–S statistic 0.1979; p value 0.0958), while response to R in the decoy treatment is either C or to a lesser extent R. These results suggest that the presence or absence of decoy greatly affects the first round of an encounter. In the later rounds, by contrast, volunteers play what seems to be noisy tit-for-tat

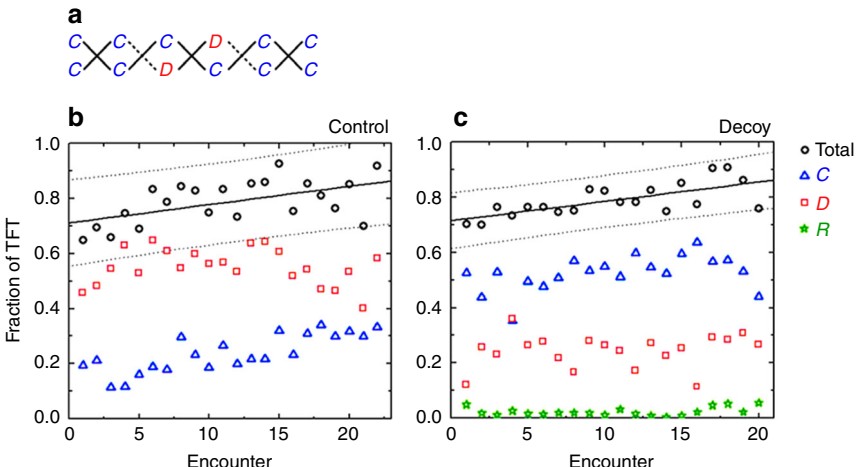

**Fig. 4** Tit-for-tat (i.e., conditional cooperation) is the prevailing strategy. **a** To calculate the fraction of tit-for-tat (TFT) actions, we looked at each encounter individually and added all instances in which C was followed by C, D by D, or—when available—R by R (solid lines). This sum was then divided by the total number of possible TFT actions (solid and dashed lines), which equals 2× (number of rounds −1). In the present example, the fraction of TFT actions is $\frac{4}{5} = 0.8$. **b** The average fraction of TFT actions in the control treatment is high from the beginning and slowly increases over the course of 20+ encounters (intercept 0.711, 95% confidence interval [0.681–0.742]; slope 0.007, 95% confidence interval [0.004–0.009]; coefficient of determination $R^2 = 0.292$). Also shown is a breakdown of how much C and D contribute to the total fraction of TFT actions. **c** The average fraction of TFT actions in decoy and control treatments is similar (intercept 0.715, 95% confidence interval [0.694–0.735]; slope 0.007, 95% confidence interval [0.005–0.009]; coefficient of determination $R^2 = 0.485$). In the decoy treatment, however, the contribution of cooperative TFT actions is considerably higher than in the control treatment. Reward R contributes very little. Solid lines represent the ordinary least squares regression, whereas dashed lines are the corresponding 95% prediction confidence intervals

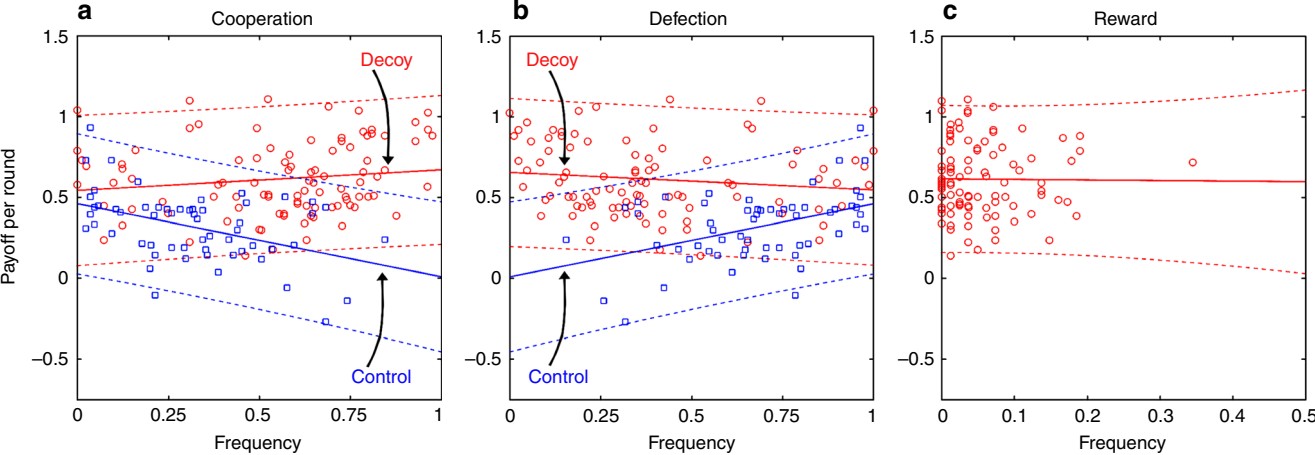

**Fig. 5** Reward improves the likelihood of success for cooperators. **a** In the control treatment, per-round payoff correlates negatively with cooperation frequency (intercept 0.459, 95% confidence interval [0.369–0.549]; slope −0.452, 95% confidence interval [−0.691 to −0.212]; coefficient of determination $R^2 = 0.197$). This negative correlation disappears in the decoy treatment (intercept 0.541, 95% confidence interval [0.431–0.650]; slope 0.128, 95% confidence interval [−0.0511 to 0.306]; coefficient of determination $R^2 = 0.0205$). In fact, the angle between the two lines is significantly positive (F-test for the treatment × frequency interaction; F-statistic 13.2; p value 0.0004), indicating that the decoy's presence improves the likelihood of success for cooperators. **b** Because frequencies of cooperation and defection in the control treatment sum to unity, regression lines for cooperation and defection in this treatment must exhibit a mirror symmetry (intercept 0.0075, 95% confidence interval [−0.164 to 0.179]; slope 0.452, 95% confidence interval [0.212–0.691]; coefficient of determination $R^2 = 0.197$; cf. panel (**a**)). Although the same symmetry need not hold in the decoy treatment due to reward, regression lines for cooperation and defection are also almost an ideal mirror image of one another (intercept 0.654, 95% confidence interval [0.575–0.733]; slope −0.109, 95% confidence interval [−0.276 to 0.0577]; coefficient of determination $R^2 = 0.0173$; cf. panel (**a**)), indicting that reward worsens the likelihood of success for defectors (F-test for the treatment × frequency interaction; F-statistic 12.9; p value 0.0004). **c** Per-round payoff shows no correlation with the frequency of reward (intercept 0.614, 95% confidence interval [0.552–0.676]; slope −0.0355, 95% confidence interval [−0.808 to 0.737]; coefficient of determination $R^2 = 0.0083$), thus explaining the mirror symmetry of regression lines for cooperation and defection in the decoy treatment. Solid lines represent the ordinary least squares regression, whereas dashed lines are the corresponding 95% prediction confidence intervals

Our interpretation of the effectiveness of reward $R$ is predicated on the correct perception of how valuable $R$ is relative to cooperation $C$. We tested this perception in additional treatments in which the payoff matrix from Eq. (1) was generalised with reward parameter $\alpha$ (Fig. 6a), whose value was increased from $\alpha = 3$ to $\alpha = 4$ to $\alpha = 5$. These increased values of $\alpha$ bring the dilemma strength of the $(D, R)$ pair first to DS = 1.5 and then DS = 1, thus making $R$ increasingly superior to $C$. Volunteers respond to changes in the relation between $R$ and $C$ as expected from the calculated dilemma strengths (Fig. 6b). The former option becomes more frequent than the latter as the distinction between the two becomes clearer. Increased cooperativeness in the decoy treatment is therefore truly attributable to a cognitive bias, specifically, the decoy effect. To account for all eventualities, we also examined how the results between treatments change due to two common confounding factors (Supplementary Note 3): gender (Supplementary Table 2) and academic background (Supplementary Table 3).

## Discussion

Returning to the question on the mechanisms underlying selection, we attempt to provide an answer by connecting several pieces of evidence. First, the results here, as well as that of a similarly structured experiment[32], were successfully recreated in computer simulations using the data on how opponents respond to each other's most recent actions (Supplementary Note 4; Supplementary Fig. 6). That most recent actions are the only ones that are truly relevant is consistent with noisy TFT described in Fig. 3. To the extent that noise cancels out, which is a first-order effect, TFT should propagate initial frequencies of cooperation $C$ and defection $D$ through time because $C$ is met with $C$ and $D$ is met with $D$. We find, however, that noise fails to cancel out completely, but rather exhibits a second-order effect in the form

of a bias toward defection (Supplementary Note 5; Supplementary Fig. 7). This second-order effect acts as if the decoy's effectiveness wears off through the course of an encounter. Fortunately, at the beginning of the very next encounter, the decoy's effectiveness is restored, causing volunteers to be more cooperative again. The overall result is that, despite the fact that volunteers correctly perceive reward $R$ as an inferior option, the initial burst of cooperativeness caused by the decoy effect is stabilised across more than 80 rounds of the game (Fig. 2).

Our study thus points to decoys as a means to elicit voluntary prosocial action across a spectrum of collective endeavours. A hypothetical example would be a small business team in which a member has fallen behind schedule. Although others may be reluctant to put extra hours to help their distressed colleague even if there is adequate overtime pay (the cooperative option), preferences may change by setting this overtime pay to decrease considerably some weeks before the deadline (the decoy option). Explained from such a psychological perspective, our results run the risk of appearing somewhat unsurprising. We therefore invite readers to contrast this explanation with an attempt at explaining why reward is a cooperation promoter solely within the bounds of evolutionary game theory. The latter explanation is unlikely to invoke cognitive biases as particularly relevant and would struggle with reward's effectiveness before anyone can actually reward anyone. Deeper implications of the results for the evolution of human cooperation are admittedly less obvious (Supplementary Discussion), but with promising research directions crystallised, we believe that maintaining an optimistic perspective is warranted.

## Methods

**Experimental Protocol and Execution.** We executed a total of 11 experimental sessions from May 2015 to April 2016 at the Behavioural Economics Lab of Yunnan University of Finance and Economics in Kunming City, China. Two

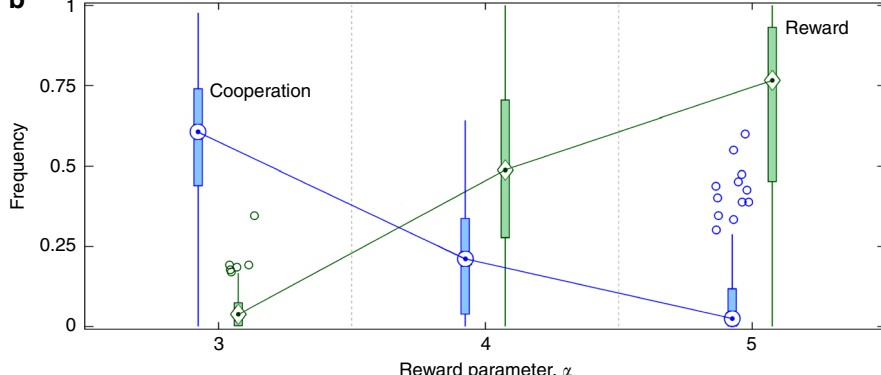

**Fig. 6** Decoy's relevance is perceived correctly. **a** Payoff matrix generalised with the reward parameter, $\alpha$. Our interpretation of the results was predicated on the correct perception of the decoy as an inferior option. To test whether this was true, we increased the reward parameter from $\alpha = 3$ to $\alpha = 4$ to $\alpha = 5$, thus decreasing the dilemma strength of the $(D, R)$ pair from DS = 3 to DS = 1.5 to DS = 1, respectively. In the process, cooperation $C$ was expected to switch places with reward $R$ as the inferior option because dilemma strength of the $(C, D)$ pair is DS = 2. **b** Our expectation is mirrored in the experimental data as the median frequency of $C$ falls from 60.5% for $\alpha = 3$ to 2.5% for $\alpha = 5$, while the median frequency of $R$ goes up from 3.7% for $\alpha = 3$ to 76.5% for $\alpha = 5$. For $\alpha = 4$, similar dilemma strengths of $(D, R)$ and $(C, D)$ pairs (DS = 1.5 vs. DS = 2, respectively) make the distinction between $C$ and $R$ somewhat opaque. With the median frequency of $C$ at 21.1% and $R$ at 48.8%, the data seem to reflect a certain degree of confusion. Despite this confusion, $R$ is more prevalent than $C$, which is in agreement with the slightly more favourable dilemma strength of the former option. The plot is a compact variant of the box-and-whisker plot in Fig. 1 and carries similar information content (medians, interquartile ranges, minimum and maximum values, and outliers if any)

sessions were dedicated to the control treatment, three to the decoy treatment, and additional six to testing whether volunteers correctly value reward $R$ relative to cooperation $C$. An average of 35.3 volunteers per session participated in approximately 82.0 rounds of the game over the course of 22.9 encounters. No volunteer was allowed to participate in more than one session. The experiment was coded using the z-Tree software[39].

We split each experimental session into three stages: preparatory, main and payout. During preparations, incoming volunteers were randomly assigned to isolated computer cubicles, where they would find instructions displayed on their computer screens, followed by a pre-game test to check the basic understanding of the rPD game. Thereafter, randomised pairing of volunteers preceded a practice pairwise encounter consisting of several rounds. In each round, paired volunteers were given 30 s to select one of the available options, and then another 30 s to examine the consequences of their selections.

The main stage of the experiment entailed a continuing sequence of randomised pairing and pairwise encounters for about 75 min. To keep track of individual success, each volunteer was endowed with an initial balance of 50 units, which changed from round to round based on decisions made and the rules in Eq. (1). Final balance, if positive, was converted into a monetary payout at a rate of ¥0.2 for 1 unit. This was supplemented with a show-up fee of ¥15, for an average payout of ¥43.3, ranging between ¥15 and ¥75.2.

In designing the experimental protocol, particular attention was paid to minimise framing effects. Terms such as "cooperation", "defection" and "reward" invariably carry positive or negative connotations which may affect behaviour during the experiment. Seeing oneself as a cooperator may boost one's self-image, whereas seeing oneself as a defector may affront this self-image. Therefore, options available to volunteers during the experimental sessions were labelled simply "1", "2" and "3" and described in terms of the effect on one's own payoff, i.e., the unilateral payoff matrix in Eq. (1).

**Ethics statement**. The experiment was approved by the Yunnan University of Finance and Economics Ethics Committee on the use of human participants in research, and carried out in accordance with all relevant guidelines. We obtained informed consent from all volunteers.

**Data availability**. The datasets generated and analysed during the current study are available in the Open Science Framework repository, https://doi.org/10.17605/OSF.IO/EHJS3[40].

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

## Acknowledgements

We are grateful to Prof. Petter Holme, Prof. Xuelong Li, Dr. Yuichi Iwasaki, Dr. Peican Zhu, Dr. Jinzhuo Liu and Dr. Chen Chu for useful discussions and Chen Shen, Yini Geng, Hao Guo, Yongjuan Ma, Danyang Jia and Yuan Li for technical help. We acknowledge support from (i) the Research Grant Programme of Inamori Foundation to M.J., (ii) the National Natural Science Foundation of China (Grant nos. 31700393 and 11671348) to L.S., (iii) the National 1000 Young Talent Plan (No. W099102), the Fundamental Research Funds for the Central Universities (no. G2017KY0001), and China Computer Federation–Tencent Open Fund (No. IAGR20170119) to Z.W. and (iv) the Japan Society for the Promotion of Science (JSPS), Grant-in-Aid for Scientific Research B to Y.I. (No. 15H04423).

## Author contributions

Z.W., M.J., J.-H.L. Y.I. and S.B. designed the experiment. Z.W. and L.S. performed the experiment. Z.W. and M.J. analysed the data. M.J. and J.-H.L. performed mathematical and numerical analyses. All authors discussed the results and wrote the manuscript.

## Additional information

**Competing interests:** The authors declare no competing interests.

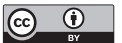

