## [Peer Review File · Nature Communications]

Reviewers' comments:

Reviewer #1 (Remarks to the Author):

Report on Jusup et al

This paper studies cooperation in indefinitely repeated Prisoners Dilemma games, where a dominated strategy is added, to elicit a "decoy effect", that is, to exploit a psychological bias demonstrated in the marketing literature that adding an inferior alternative to a choice set has consequences for choices that are inconsistent with standard views on rationality, that is, can lead to choices that people do not make in the absence of a decoy effect. The results of this study find evidence for such a decoy effect, in particular, that its presence increases cooperation compared to a control treatment without a decoy. The effect seems to operate primarily with first round cooperation; conditional reactions to previous round choices are quite similar between treatments.

The experiments are done competently and the analysis is interesting (although I have some comments on the statistical analysis, see below). The paper is well written.

From a psychological viewpoint, the results make sense, but are also not very surprising. Adding an inferior alternative can make people choose the alternative it resembles most, in the present case, it makes cooperation "more attractive". Compared to the control treatment, this effect works by shifting initial cooperativeness, to which people then respond by 'tit-for-tat' in the same way as in the control treatment. Because it is a specific cognitive bias that may work in this stylized setting, but may be less general across settings (see comment 2 below) I am less sure about the significance of the decoy effect for understanding the evolution of cooperation.

Detailed comments

- 1) The first paragraph is way too general and does not really set the scene for what is to follow later on. As is currently written, I think the paper could just start with the current second paragraph. Even better, maybe, would be to introduce the decoy effect and the related literature more prominently.
- 2) The decoy effect receives only one citation (ref. 12). It should also be acknowledged that there is a discussion about the applicability and replicability of the decoy effect (known also as asymmetric dominance effect, or attraction effect). See doi: 10.1509/jmr.12.0061, doi: 10.1509/jmr.14.0020, doi: 10.1509/jmr.14.0093, doi: 10.1509/jmr.14.0208.
- 3) Third paragraph: the nature of the decoy option (adding a dominated strategy) should be mentioned here already. It would enhance the reader's understanding of what is to follow.
- 4) The continuation probability, expected length, and nature of encounters should also be mentioned in the description of the experimental design. This information is much more important than when and where the experiments were conducted. Such information is best placed in the methods section.
- 5) The supplementary materials should include the exacts translated texts of the experimental instructions given to the participants.
- 6) Fig. 1: What are the independent observations entering the Mann-Whitney tests? Also, why one-tailed tests?
- 7) Figs. 2 & 3: Same comment: what are the independent observations? The last line of the caption text is a bit garbled.
- 8) Fig. 2: This averages over all encounters. In how many encounters did the frequency of C decrease over interactions?
- 9) Fig. 3 is quite interesting and insightful. I also wondered how first-round cooperation rates changed over encounters.
- 10) Fig. 4: Explain the statistical model used to produce panel D.
- 11) Fig. 5: Same comment as Fig. 4. Are the dashed lines confidence intervals?
- 12) Page 5, paragraph "In evolutionary game theory ...": I don't find it very puzzling that the

authors don't find that "cooperativeness leads to success in the decoy treatment". The games are very short in expectation (as the participants probably understood at least intuitively). Also, in response to the last couple of sentences in that paragraph, long games and some targeted punishment might be necessary to make cooperation beneficial (doi:10.1126/science.1164744).

13) Fig. 6: The colors chosen for Cooperation and Reward are hardly distinguishable.

14) Page 5, penultimate line: I don't understand the claim that "the relation between R and C [is] precisely as rationality dictates". Adding a constant to all payoffs of R does not change the fact that R is dominated by both C and D and rational players should therefore always ignore R. I guess the authors have the repeated games in mind, and if so, should say so, or modify the claim about rationality. The confusion continues in the next sentence where it says that "the former option becomes dominant over the latter ...". I guess, here "dominant" means "more frequent", and does not refer to dominance according to game-theoretic rationality.

15) Middle of page 6: "... paired with ..." rather than "pared with"?

16) Discussion: given the state of the literature on the decoy effect, which paints a less than robust picture of its relevance in many real-world settings, I am somewhat sceptical about the practical applicability of this research. If, e.g., in an organizational context, choice architects can add irrelevant alternatives to the choice sets, why can't they deal with the primary social dilemma?

Reviewer #2 (Remarks to the Author):

In this paper the authors attempt to apply a context effect that is typically studied in preference formation in a single player format to a two player repeated Prisoner's dilemma game. As such, I think this is an important extension of behavioral economics into game theory that is, as far as I am aware, novel. From my experience with decoy effects, I see no reason to believe that this manipulation would not work, and the pattern of results reported in the research are similar to what effect I would expect an Asymmetrically dominated decoy targeting the Cooperation option would have. That said, I have some major reservations about the paper that I would like to see addressed. Let me state that while I am an expert in Decoy effects, I am not an expert in game theory, so my comments focus on the decoy side of things mostly.

1. In behavioral economics, "Decoy" is a generic term that can refer to any of a number of methods of placing a third alternative in a choice set relative to its target. Generally, papers studying decoy effects make clear which kind of decoy they are using, because they all have somewhat different effects and are often explained by different cognitive models. This paper does not make clear what kind of decoy it is using. I think they are trying to use an Asymmetrically Dominated Decoy (AD), often referred to as the Attraction effect, but I'm not convinced that their Reward option is dominated by the decoy. Generally AD decoys share one feature of the target and are inferior on a second. For example I think the following would be a true AD decoy in this setting...

AD (-1 1)

As it is truly inferior to the cooperation option, in that it costs the player the same as C but provides less to the Foe. Thus, it truly is inferior and should never be chosen. The reward used in this paper is interesting, but at least at face value not inferior. It does cost more, but it should also reward more if both players choose it. It is inferior in final payout (See #2 for a question about this), but not necessary inferior in the sense of how decoys are used in BE, as one could still justify choosing it because it does give the other player more. In that regard, the R decoy actually looks more like a compromise decoy, in that it extends the range of values on both attributes of the target. Either way, including this AD would result matrix A1 in the attachment...

Either way, the authors need to spend more time explaining how their decoy fits in with the extant decoy literature and which specific type of decoy (AD, Compromise, or other) they are trying to use. Any third, inferior option is not necessarily a decoy. In fact, compromise decoys sometimes are not inferior options at all! Also, in this situation, I think a decoy needs to be inferior both in comparison to the other options at the time of choice and in terms of final outcomes, as otherwise choosing it may be a rational decision.

2. Related to #1, I'm not sure that the result matrix reported in equation 1 for the R decoy is correct. When I calculate it out, I get very different results for when the player chooses R. My calculations result in the following payout structure A2 in the attachment...

Again, I'm not an expert here, so I may be incorrect with this, but I would appreciate an expert double checking the math here. If this matrix is true, including R actually improves the option of choosing C (compared to the two option control) because the player gets a positive outcome based on a play of C AND R now by the other player. Choosing R, however, would not make much sense over time and would be inferior to both other options, or equal if both players always choose R. Thus, I question whether the choice of C in this paper is a bias or a rational response to the payout structure.

So, this is a pretty major thing that needs to be checked out. I may be wrong, but I'd like someone else to verify this. If I'm right, then the Reward option in this paper is not a decoy, and the player behavior is rational.

In the end, this paper needs to connect better to the decoy literature as well as verify it's payoff structure before I could recommend publication. I think the idea is novel and potentially very interesting- I'm just not sure this is a decoy effect, and not just a rational response to the matrix I provided in #2 above where C is still interior to D, but its somewhat LESS inferior with R included than without. Again- I'm not an expert in game theory here, so I'm cautious to make such a statement, and I'll apologize in advance if I'm incorrect. I'd like to see a replication with the AD decoy suggested above either way.

Jonathan Pettibone
Dept. of Psychology
Professor, Southern Illinois University Edwardsville

A1. Proposed AD payoff matrix

	C	D	AD	
C		1	-2	0
D		3	0	2
AD		1	-2	0

A2. Recalculated R payoff matrix.

	C	D	R	
C		1	-2	2
D		3	0	4
R		0	-3	1

We thank the reviewers for thoroughly checking the manuscript and providing useful comments. The final manuscript will certainly be better thanks to their inputs. Here, we list several conventions which should help with readability of the responses below.

- All reviewer comments are displayed in *black italics*.
- All our responses are displayed in **red-brown upright letters**.
- Sentences indicating changes to the manuscript are underlined.

Reviewer #1:

Comment: *This paper studies cooperation in indefinitely repeated Prisoners Dilemma games, where a dominated strategy is added, to elicit a “decoy effect”, that is, to exploit a psychological bias demonstrated in the marketing literature that adding an inferior alternative to a choice set has consequences for choices that are inconsistent with standard views on rationality, that is, can lead to choices that people do not make in the absence of a decoy effect. The results of this study find evidence for such a decoy effect, in particular, that its presence increases cooperation compared to a control treatment without a decoy. The effect seems to operate primarily with first round cooperation; conditional reactions to previous round choices are quite similar between treatments.*

The experiments are done competently and the analysis is interesting (although I have some comments on the statistical analysis, see below). The paper is well written.

Reply: Thank you for your positive assessment of the manuscript. We feel encouraged to keep up the good work. Accordingly, we have responded to all the reviewer's comments in great detail, which will hopefully prove sufficient to secure the acceptance of our work by the journal.

Comment: *From a psychological viewpoint, the results make sense, but are also not very surprising. Adding an inferior alternative can make people choose the alternative it resembles most, in the present case, it makes cooperation “more attractive”. Compared to the control treatment, this effect works by shifting initial cooperativeness, to which people then respond by “tit-for-tat” in the same way as in the control treatment. Because it is a specific cognitive bias that may work in this stylized setting, but may be less general across settings (see comment 2 below) I am less sure about the significance of the decoy effect for understanding the evolution of cooperation.*

Reply: We agree with the reviewer that the results may not be overly surprising *once they are explained from a psychological perspective*. However, we tend to see this as a sign of a good explanation; the results are placed in an easy-to-understand---and thus less surprising---context. By thinking solely within the bounds of evolutionary game theory to explain the cooperation-promoting effect of reward, one is unlikely to ever invoke cognitive biases as particularly relevant. Such an explanation would almost certainly struggle with the effectiveness of reward before anyone can actually reward anyone. We explicitly stated this in the last paragraph of the manuscript.

As for the significance of the results for the evolution of cooperation, we are influenced by recent developments showing that what appear as cognitive biases in a laboratory setting may have been beneficial for survival in our ancestral environments, e.g., <10.1016/j.tics.2013.12.012>. We do admit in Supplementary Discussion that connections made in this context are somewhat speculative, but this is precisely the reason why the corresponding lines were placed in the supplementary file in the first place. We also believe that our results warrant further research, as stated in the last paragraph of the manuscript.

Comment: 1) *The first paragraph is way too general and does not really set the scene for what is to follow later on. As is currently written, I think the paper could just start with the current second paragraph. Even better, maybe, would be to introduce the decoy effect and the related literature more prominently.*

Reply: We followed the reviewer's advice and introduced the decoy effect as suggested. We would also like to emphasise that our intention with this first paragraph was to better serve the journal's general audience, which may not be fully informed of the relevant developments in the field. With the direct mention of the decoy effect, we do feel that our intention is realised in a better way.

Comment: 2) *The decoy effect receives only one citation (ref. 12). It should also be acknowledged that there is a discussion about the applicability and replicability of the decoy effect (known also as asymmetric dominance effect, or attraction effect). See doi:10.1509/jmr.12.0061, doi:10.1509/jmr.14.0020, doi:10.1509/jmr.14.0093, doi:10.1509/jmr.14.0208.*

Reply: We followed the reviewer's advice fully. Please also see the previous reply. We are especially grateful for useful references.

Comment: 3) *Third paragraph: the nature of the decoy option (adding a dominated strategy) should be mentioned here already. It would enhance the reader's understanding of what is to follow.*

Reply: We agree and believe that this has been achieved with modifications to the text in response to the two preceding comments, as well as the corresponding adjustments to the third paragraph itself.

Comment: 4) *The continuation probability, expected length, and nature of encounters should also be mentioned in the description of the experimental design. This information is much more important than when and where the experiments were conducted. Such information is best placed in the methods section.*

Reply: We modified the methodological paragraph spanning pages 3 and 4, as well as the Methods section, to comply with the reviewer's suggestion.

Comment: 5) *The supplementary materials should include the exacts translated texts of the experimental instructions given to the participants.*

Reply: We did exactly as instructed by the reviewer.

Comment: 6) *Fig. 1: What are the independent observations entering the Mann-Whitney tests? Also, why one-tailed tests?*

Reply: When testing, for example, whether the median cooperation is different between the two treatments, we counted for each volunteer the number of cooperative actions taken, divided this count by the number of rounds played, and then inserted the obtained values into the Mann-Whitney test. Because of this question by the reviewer, we extended the explanation in the caption of Fig. 1. Furthermore, the reason for a one-tailed test is that we hypothesised before executing the experiment that the decoy's presence would either increase cooperation or do nothing at all---a one-sided effect.

Comment: 7) Figs. 2 & 3: Same comment: what are the independent observations? The last line of the caption text is a bit garbled.

Reply: In Fig. 2, we counted how many volunteers chose a particular action and divided these counts by the total number of volunteers playing. When mentioning a “garbled sentence”, we were not sure whether the reviewer was referring to the caption of Fig. 2 or 3. We assumed it was Fig. 2 and wrote a, hopefully, more understandable sentence. In Fig. 3, we performed the Mann-Whitney test on the values obtained analogously to the ones in Fig. 1. We clarified this in the caption of Fig. 3.

Comment: 8) Fig. 2: This averages over all encounters. In how many encounters did the frequency of C decrease over interactions?

Reply: Thank you for asking this interesting question as it helped us to uncover an important second-order effect lurking in the data. Before going into details, however, we first wish to give a straight answer to the reviewer's question. Out of 53 encounters lasting >1 round, only 3 ended up with a higher cooperation frequency than at the beginning of the encounter. In the remaining 50 encounters, the cooperation frequency at the end was lower than at the beginning.

Having drawn our attention, we examined within-encounter cooperativeness more closely and reported our findings in a separate section of Supplementary Material. The reason why we originally discounted the importance of this type of analysis is our previous experience described here <doi:10.1126/sciadv.1601444>, where we did not find any significant trend in within-encounter cooperation frequency.

The analysis prompted by this question revealed that the noise in “noisy” tit-for-tat does not entirely cancel out. There is a bias towards defection with a non-negligible impact on the overall cooperativeness: if the noise cancelled out (a first-order effect), we would expect the 1st-round cooperation frequency of $\approx 74\%$ to approximate the average cooperation frequency throughout the experiment. This, however, is not the case as reported in Fig. 2, wherein the average cooperation frequency is $\approx 54\%$, meaning that the observed bias in the noise (a second-order effect) has a meaningful impact on the results. What helps the cooperation to persist then---aside from tit-for-tat---is that the decoy's effectiveness is restored at the beginning of every encounter although it gradually wears off within an encounter. In the second-to-last paragraph of the main text, we therefore write: “To the extent that noise cancels out, which is a first-order effect, tit-for-tat should propagate initial frequencies of cooperation C and defection D through time because C is met with C and D is met with D . We find, however, that the noise fails to cancel out completely, but rather exhibits a second-order effect in the form of a bias towards defection (SM, Within-encounter cooperativeness). This second-order effect acts as if the decoy's effectiveness wears off through the course of an encounter. Fortunately, at the beginning of the very next encounter, the decoy's effectiveness is restored, causing volunteers to be more cooperative again.”

Comment: 9) Fig. 3 is quite interesting and insightful. I also wondered how first-round cooperation rates changed over encounters.

Reply: We were indeed hoping that reviewers would find Fig. 3 informative. Thank you. As for how the 1st-round cooperation frequency changes over encounters, we provide a figure below (shown is the case $\alpha=3$). This is just to confirm what actually Fig. 2 already implies. Because the overall cooperation frequency is stable and volunteers play tit-for-tat, the 1st-

round cooperation frequency must also stay relatively stable; if it did not, then tit-for-tat would cause the overall cooperation frequency to go down as well.

Comment: 10) Fig. 4: Explain the statistical model used to produce panel D.

Reply: We added an explanation to the caption of Fig. 4 as suggested by the reviewer.

Comment: 11) Fig. 5: Same comment as Fig. 4. Are the dashed lines confidence intervals?

Reply: We added an explanation to the caption of Fig. 5 as suggested by the reviewer.

Comment: 12) Page 5, paragraph “In evolutionary game theory ...”: I don’t find it very puzzling that the authors don’t find that “cooperativeness leads to success in the decoy treatment”. The games are very short in expectation (as the participants probably understood at least intuitively). Also, in response to the last couple of sentences in that paragraph, long games and some targeted punishment might be necessary to make cooperation beneficial (doi:10.1126/science.1164744).

Reply: Perhaps the reviewer would better appreciate our viewpoint in the context of the results here <doi:10.1126/sciadv.1601444>. In this study, conducted in a similar setting as the present one, the treatment leading to increased cooperation also leads to a significant positive correlation between payoff and cooperation frequency. We see the same tendency here (the angle between the two lines in Fig. 5 is >0), but the improvement is not quite sufficient for a significant positive correlation.

We are grateful for the suggested reference which paints a slightly different picture about the role of punishment. This reference is now included in the manuscript for interested readers to check.

Comment: 13) Fig. 6: The colors chosen for Cooperation and Reward are hardly distinguishable.

Reply: Throughout the manuscript, we tried to stick to the three basic colours---red, green, and blue---wherever possible. Perhaps the best example is Fig. 3. We also tried to be consistent by always representing cooperation with blue and reward with green. We

suggest that for the sake of consistency, colours in Fig. 6 remain as is. However, we did try to improve readability by labelling cooperation vs. reward more clearly.

14) Page 5, penultimate line: I don't understand the claim that "the relation between R and C [is] precisely as rationality dictates". Adding a constant to all payoffs of R does not change the fact that R is dominated by both C and D and rational players should therefore always ignore R. I guess the authors have the repeated games in mind, and if so, should say so, or modify the claim about rationality. The confusion continues in the next sentence where it says that "the former option becomes dominant over the latter ...". I guess, here "dominant" means "more frequent", and does not refer to dominance according to game-theoretic rationality.

Reply: We apologise for the confusing language. Especially the use of word "dominant" here goes against its strict, game-theoretic meaning. The reviewer is absolutely right and we corrected the text as suggested.

Comment: 15) Middle of page 6: "... paired with ..." rather than "pared with"?

Reply: This is a mistake. Perhaps the sentence makes most sense if "pared" is replaced with "met". Thank you for noticing.

16) Discussion: given the state of the literature on the decoy effect, which paints a less than robust picture of its relevance in many real-world settings, I am somewhat sceptical about the practical applicability of this research. If, e.g., in an organizational context, choice architects can add irrelevant alternatives to the choice sets, why can't they deal with the primary social dilemma?

Reply: We appreciate the reviewer's point that transferring the results from a somewhat artificial domain, such as a computer lab, to a more practical domain, such as a business organisation, may not be an easy task. However, we do find that the decoy is repeatedly effective at the beginning of each new encounter even if its effectiveness gradually wears off within-encounter. A practical implication is that encountering a potential gift-bearer puts us in a more cooperative mood as argued in Supplementary Discussion. This implication also yields a hypothesis testable in a more practical domain. *We are, in fact, contemplating an experiment designed to test such a hypothesis.*

* * *

Reviewer #2:

Comment: *In this paper the authors attempt to apply a context effect that is typically studied in preference formation in a single player format to a two player repeated Prisoner's dilemma game. As such, I think this is an important extension of behavioral economics into game theory that is, as far as I am aware, novel. From my experience with decoy effects, I see no reason to believe that this manipulation would not work, and the pattern of results reported in the research are similar to what effect I would expect an Asymmetrically dominated decoy targeting the Cooperation option would have. That said, I have some major reservations about the paper that I would like to see addressed. Let me state that while I am in expert in Decoy effects, I am not an expert in game theory, so my comments focus on the decoy side of things mostly.*

Reply: We are glad that the reviewer sees our contribution as novel. This is an important

source of motivation for us to continue working on this class of problems. We also find it encouraging that the reviewer's experience is in line with ours in the sense that there are no reasons why a decoy targeting cooperation would not work, which balances a slightly more sceptical view of reviewer #1. As for the reviewer's concerns, we earnestly tried to address them to everyone's satisfaction. Please see our replies below.

Comment: 1. In behavioral economics, "Decoy" is a generic term that can refer to any of a number of methods of placing a third alternative in a choice set relative to its target. Generally, papers studying decoy effects make clear which kind of decoy they are using, because they all have somewhat different effects and are often explained by different cognitive models. This paper does not make clear what kind of decoy it is using. I think they are trying to use an Asymmetrically Dominated Decoy (AD), often referred to as the Attraction effect, but I'm not convinced that their Reward option is dominated by the decoy. Generally AD decoys share one feature of the target and are inferior on a second. For example I think the following would be a true AD decoy in this setting...

AD (-1 1)

As it is truly inferior to the cooperation option, in that it costs the player the same as C but provides less to the Foe. Thus, it truly is inferior and should never be chosen. The reward used in this paper is interesting, but at least at face value not inferior. It does cost more, but it should also reward more if both players choose it. It is inferior in final payout (See #2 for a question about this), but not necessary inferior in the sense of how decoys are used in BE, as one could still justify choosing it because it does give the other player more. In that regard, the R decoy actually looks more like a compromise decoy, in that it extends the range of values on both attributes of the target. Either way, including this AD would result matrix A1 in the attachment...

Either way, the authors need to spend more time explaining how their decoy fits in with the extant decoy literature and which specific type of decoy (AD, Compromise, or other) they are trying to use. Any third, inferior option is not necessarily a decoy. In fact, compromise decoys sometimes are not inferior options at all! Also, in this situation, I think a decoy needs to be inferior both in comparison to the other options at the time of choice and in terms of final outcomes, as otherwise choosing it may be a rational decision.

Reply: This is an extremely important comment because it illustrates how combining concepts from two scientific fields can cause considerable misunderstandings. We are very grateful to the reviewer for bringing this issue up, and for providing a very detailed explanation.

The key to resolving this matter, we believe, lies in carefully examining the reviewer's definition: "A[symmetrically] D[ominated] decoys share one feature of the target and are inferior on a second", and what this definition means in the context of evolutionary game theory---the only context relevant to our study. Namely, in evolutionary game theory, whether one cooperative action dominates over the other is determined by how the two actions fare against exploitation by defectors. This is where the concept of Dilemma Strength (DS) is useful because a higher value of DS means more severe exploitation (as explained at length in SM, Dilemma Strength). In our study, (i) cooperation C and reward R share the feature of being cooperative actions, but (ii) R (DS=3) is less resilient to exploitation by defectors than C (DS=2), and thus an inferior action. The choice we made, therefore, directly satisfies the reviewer's definition---points (i) and (ii)---and it does so in the context relevant to evolutionary game theory.

To avoid future misunderstandings, and to better connect our study to the literature on the decoy effect (also pointed out by reviewer #1), we substantially expanded the first

paragraph of the manuscript, added several key references, and included the reviewer's definition of the decoy there. Furthermore, in the paragraph introducing DS, we explicitly stated that our decoy satisfies this definition.

For completeness, let us briefly examine the payoff (AD) proposed by the reviewer. With this payoff, the dilemma strength of the (AD, D) pair is $+\infty$, indicating an instance of PD which favours defection to the maximum extent possible. In a game-theoretic sense, therefore, AD is not just inferior to C , but also to R . One could intuitively imagine DS as a measure of distance (or distinction) between C, R , and AD on the diagonal displayed in the coordinate system in Fig. S3. If volunteers sense the distance between C and R ---which is something they demonstrate by ignoring R in favour of C (see also results in Fig. 6)---there is little doubt that they would refuse using AD when C is available. In fact, because the distinction between C and AD is much clearer than between C and R , it is highly likely that AD would work less well than R as a decoy.

One important aspect of AD unmentioned so far is that this payoff cannot possibly be interpreted as reward. However, evolutionary biologists have long hypothesised that reward is an important, if not key, factor in driving the evolution of cooperation. When selecting a payoff structure for action R , we had to keep in mind the relevance of this choice for the field of evolutionary game theory, and more widely, evolutionary biology.

Comment: 2. *Related to #1, I'm not sure that the result matrix reported in equation 1 for the R decoy is correct. When I calculate it out, I get very different results for when the player chooses R . My calculations result in the following payout structure A2 in the attachment...*

Again, I'm not an expert here, so I may be incorrect with this, but I would appreciate an expert double checking the math here. If this matrix is true, including R actually improves the option of choosing C (compared to the two option control) because the player gets a positive outcome based on a play of C AND R now by the other player. Choosing R , however, would not make much sense over time and would be inferior to both other options, or equal if both players always choose R . Thus, I question whether the choice of C in this paper is a bias or a rational response to the payout structure.

So, this is a pretty major thing that needs to be checked out. I may be wrong, but I'd like someone else to verify this. If I'm right, then the Reward option in this paper is not a decoy, and the player behavior is rational.

Reply: The reviewer is absolutely right about there being a mistake in Eq. (1), despite having the correct generalised payoff matrix displayed as a part of Fig. 6. We made the necessary correction in the manuscript. We initially intended for Eq. (1) to display the generalised payoff matrix with parameter α as in Fig. 6, but to streamline the discussion, we finally decided to simplify Eq. (1) by inserting value $\alpha=3$ into the matrix. The first author unfortunately forgot to make this insertion and nobody else noticed. We thank you very much for spotting our, somewhat embarrassing, mistake!

As for the concern that action R is not a decoy, we believe that this has been alleviated in the previous reply.

Comment: In the end, this paper needs to connect better to the decoy literature as well as verify it's payoff structure before I could recommend publication. I think the idea is novel and potentially very interesting- I'm just not sure this is a decoy effect, and not just a rational response to the matrix I provided in #2 above where C is still interior to D , but its somewhat LESS inferior with R included than without. Again- I'm not an expert in game theory here, so I'm cautious to make such a statement, and I'll apologize in advance if I'm

incorrect. I'd like to see a replication with the AD decoy suggested above either way.

Reply: This comment summarises the reviewer's previous comments. In response, we will summarise our replies:

- We believe that the extended first paragraph now provides a much stronger connection to the decoy literature.
- As explained, superiority of one cooperative action over the other is measured in how they resist exploitation by defectors. Dilemma strength is an intuitive indicator of this resistance / resilience (think of the position on the diagonal in Fig. S3). In our study, *C* and *R* are both cooperative (a shared feature), but *R* is less resilient to defection than (i.e., inferior to) *C*. Reward *R* is therefore a valid decoy.
- *AD* is not only inferior to *C*, but also to *R*. In fact, *AD* is as removed from *C* as possible, which suggests that *AD* would work less well as decoy than *R*. Furthermore, *AD* cannot be interpreted as reward; however, shedding light on the possible role of reward in the evolution of cooperation is what in large part motivated this research.

In the end, we would like to express our gratitude to the reviewer for a positive evaluation, challenging and useful comments, and for spotting the mistake in Eq. (1), despite the correct generalised payoff matrix being displayed in Fig. 6. We feel that all this made the revised manuscript considerably better.

Reviewers' comments:

Reviewer #1 (Remarks to the Author):

The authors did a good job in revising this paper. As a result I think the manuscript is improved. I have some further comments on the manuscript.

- 1) P. 3, middle para and equation 1: I am slightly confused by the claim that the "... decoy option – always dominated by cooperation, but not by defection in the limit of a fully cooperative population ..." given the payoff matrix of eqn 1, according to which C and D dominate R. I think this should be explained better, because otherwise the claim made on p. 3 and eqn 1 look contradictory.
- 2) P. 4, 3rd/4th line from above: shouldn't it say "... at the opponent's expense of one unit" (according to equation 1)?
- 3) P.5: The concept of dilemma strength (DS) is quite important for understanding the rest of the paper and therefore DS deserves being explained in the main text, not just the SI. Actually, I don't understand the argument made in the sentence "We find from Eq. (1), including the new couple of sentences.
- 4) P. 5, sentence "Rationality dictates ...": I don't understand this sentence. Because D dominates both R and C, rationality dictates that players only play D. Please amend/clarify.
- 5) I am not entirely happy with the answers to my question on statistically independent observations. I now understand better how averages are formed, but the tests seem to assume that people are not influenced by the experience of previous interactions; however, any spillover effects from experience make observations interdependent. Thus, I think, strictly speaking only sessions are truly independent observations and the p-values reported are "too low" because they rest on an inflated number of independent observations.

Reviewer #2 (Remarks to the Author):

First, let me apologize for the lateness of my review. I've reviewed the authors comments and gone over the changes to the manuscript, and I'm satisfied in their responses. They now do a much better job in fitting into the decoy literature, and I appropriate learning more about the focus upon dilemma strength.

I'm still not sure that this focus provides for a completely natural analog to the single player games that decoys are usually tested in. R is certainly an inferior option, but the focus on the outcome, not the initial choice set, makes comparisons more difficult. In fact, I wonder if in this use, your R is actually similar to a symmetrically dominated decoy, or RF decoy from Wedell and Pettibone (1996), in that its fully dominated by both options. There's no dimension that you can reduce the comparison between R and D to in which a choice of R is better, and its the same with C. In the AD situation, the competitor is still superior to the decoy on a single dimension. I'll leave that determination up to the authors.

Overall, I'm happy with the improvement in the attempt to reconcile the two sets of literature.

I support publication as is, after consideration of my comments.

We thank the reviewers for thoroughly checking the manuscript once again. As before, the following conventions should help with readability of the responses below.

- All reviewer comments are displayed in *black italics*.
- All our responses are displayed in **red-brown upright letters**.
- Sentences indicating changes to the manuscript are underlined.

Reviewer #1:

Comment: *The authors did a good job in revising this paper. As a result I think the manuscript is improved. I have some further comments on the manuscript.*

Reply: Thank you for positively assessing the manuscript revised in response to your comments. This means a lot.

As for the reviewer's additional concerns, we spare no effort to address them in a mutually satisfactory way, but with the quality of the manuscript in mind above all else.

Comment: *P. 3, middle para and equation 1: I am slightly confused by the claim that the "... decoy option – always dominated by cooperation, but not by defection in the limit of a fully cooperative population ..." given the payoff matrix of eqn 1, according to which C and D dominate R. I think this should be explained better, because otherwise the claim made on p. 3 and eqn 1 look contradictory.*

Reply: By making this statement somewhat prematurely in the text, we inadvertently conflated the game-theoretic meaning of "dominance" with its meaning in the context of cognitive biases. We decided to delete the statement and instead provide a more detailed explanation in the two paragraph beginning at the end of page 4, which are entirely devoted to explaining relationships between actions.

The reviewer correctly interprets Eq. (1) in a game-theoretic sense: action *D* indeed dominates over action *C*, which in turn dominates over action *R*. This is seen from the bilateral payoff matrix in which payoffs associated with *D* (*C*) are higher than the corresponding payoffs associated with *C* (*R*). If we were interested in *single-shot games* only, the discussion would end here. In *repeated games*, however, matters get more complicated.

First, despite its dominance, defection *D* is not necessarily the only evolutionarily stable strategy (ESS) in *repeated games*; cooperation *C* can also be an ESS provided there is enough protection against exploitation by *D*. To drive this point home, we write: "The effect of repetitions, by contrast, is to potentially change the nature of the dilemma³⁷, at least in a probabilistic sense (SM, Game repetitions). This means that instead of having defection *D* as the only evolutionarily stable strategy (ESS), cooperation *C* may also become an ESS." We provide additional support for this well-known result by citing an appropriate reference (Ref. 37) and explaining the effect of game repetitions in the supporting materials (SM, Game repetitions).

Second, under which conditions are cooperators sufficiently protected against exploitation by defectors? To explain, we write: "For cooperators to prevail, their initial fraction x_C^* must be sufficiently high to provide protection against excessive exploitation by defectors. This initial fraction is given by the dilemma strength parameter, DS , such that $x_C^*=q DS/(1-q)$, where q is the game termination probability. The higher the value of DS , the higher the initial fraction of cooperators must be for them to prevail."

Relationship $x_C^*=q DS/(1-q)$ illustrates that *repeated games* are, in fact, characterised by the nature of the dilemma (parameter DS) and game repetitions (parameter q). However,

we set $q=25\%$ throughout the manuscript, leaving only parameter DS to define the rPD game. Because both cooperation C and reward R are cooperative actions, the one with a higher DS value (calculated in conjunction with defection D), is the one that offers less protection against defectors. In other words, the action with the higher DS value is inferior (or dominated in the sense of the literature on cognitive biases). Accordingly, we write: “To apply the concept of dilemma strength for our purpose, we recognise from Eq. (1) that R is another form of cooperation, more costly than C , but also more beneficial for the opponent, and we look at which of the two actions better protects cooperators from excessive exploitation by defector. Rationality dictates ignoring the form of cooperation that offers less protection, meaning that rational opponents should end up playing a 2×2 game consisting of either the (C, D) pair or the (D, R) pair. We find from Eq. (1) that for these pairs $DS=2$ and $DS=3$, respectively. A higher value of DS for pair (D, R) points to a less favourable dilemma and forces a conclusion that R is inferior to C .”

In summary, the hierarchy of actions in *repeated games* is determined by parameters DS (reflecting the nature of the dilemma) and q (reflecting game repetitions). Defection D may be dominant in *single-shot games*, but in repeated games, cooperation C may prevail over D given the sufficient fraction of cooperating players. If there are two cooperative actions (e.g., reward R in addition to C), which one is inferior also depends on the values of parameters DS and q .

We believe that the above explanation is relevant not only here, but also to some of the comments below, which is why we refer back to it when necessary.

Comment: *P. 4, 3rd/4th line from above: shouldn't it say "... at the opponent's expense of one unit" (according to equation 1)?*

Reply: Thank you for spotting this mistake. Corrected!

Comment: *The concept of dilemma strength (DS) is quite important for understanding the rest of the paper and therefore DS deserves being explained in the main text, not just the SI. Actually, I don't understand the argument made in the sentence "We find from Eq. (1), including the new couple of sentences.*

Reply: We absolutely agree with the reviewer that the concept of dilemma strength is important. To address the reviewer's concern expressed here, as well as the concerns from a comment above, we considerably expanded the explanation of key concepts beginning with the last paragraph on page 4 and continuing in the next paragraph on page 5 (previously only one paragraph). We also try to build the reader's intuition by stating: "How do we generally determine which action choice is worse and which is better? If our interest was solely in single-shot games, then Eq. (1) would indicate that—in a game-theoretic sense—defection D dominates over cooperation C , which in turn dominates over reward R . This is seen from the bilateral payoff matrix in which payoffs associated with D (C) are higher than the corresponding payoffs associated with C (R). Unfortunately, game repetitions complicate matters, forcing us to consider (i) the nature of the social dilemma and (ii) the effect of repetitions. The nature of the dilemma is distilled in the concept of dilemma strength^{25, 36}, whereby the stronger the dilemma, the easier it gets for defectors to exploit cooperators (SM, Dilemma strength)." We accompany this text not only with a whole section in the supporting materials, but also with appropriate references in which an interested reader can find all the details on this concept. However, we would like the reviewer to reconsider the request to further expand the discussion on dilemma strength in the main text because there are other equally important concepts (e.g., game repetitions). It would be somewhat misleading to create an impression that these other concepts play a

less important role. On the other hand, we feel that transferring most of the contents from two sections in the supporting material (in order to fully explain the role of both dilemma strength and game repetitions) to the main text would take away the attention from the main results of the study.

As for the reviewer's remark on understanding the sentence beginning with "We find from Eq. (1)[...]", we believe that the new expanded explanation of the key concepts, including dilemma strength, will help alleviate the problem. In particular, we emphasise the fact that a higher dilemma strength of the (D, R) pair than the (C, D) pair means that reward R offers less protection against defection than cooperation C (at least when $\alpha=3$). Less protection against defection makes R inferior to C , which is the reason why R should be ignored in the decoy treatment.

Comment: *P. 5, sentence "Rationality dictates ...": I don't understand this sentence. Because D dominates both R and C , rationality dictates that players only play D . Please amend/clarify.*

Reply: We did exactly as the reviewer requested by explaining that it is rational to choose (between C and R) whichever cooperative action offers more protection against defectors. Please see also our replies above.

Comment: *I am not entirely happy with the answers to my question on statistically independent observations. I now understand better how averages are formed, but the tests seem to assume that people are not influenced by the experience of previous interactions; however, any spillover effects from experience make observations interdependent. Thus, I think, strictly speaking only sessions are truly independent observations and the p -values reported are "too low" because they rest on an inflated number of independent observations.*

Reply: We first wish to apologise for focusing solely on definitions in our original response to the reviewer's comment. Here, we try to amend the problem by carefully revisiting the concern raised by the reviewer. To this end, we performed numerical simulations whereby we checked whether this concern would materialise or not. *For the results, please see the figure below*. We furthermore offer a step-by-step explanation as to why we obtained these particular results.

We ran the mentioned simulations using the same code as in SM, Computer simulations in order to recreate the experiment many times. These recreations roughly, but satisfactorily approximate the complexities of human behaviour displayed throughout sessions of the experiment. We then used the results to run the statistical test from Fig. 1 multiple times and obtain an empirical distribution of the test statistic. Finally, we compared this empirical distribution with the distribution theoretically assumed in the test (for large samples, this is the standardised normal distribution). A key point is that, if the reviewer's concern were true, the empirical distribution would deviate from the theoretically assumed one, thus pointing to a problem with p -values in the manuscript. However, because empirical and theoretical distributions match closely, calculating p -values by means of the theoretical distribution is justified. Here follows an explanation why this result may have been expected.

The reviewer states that: "strictly speaking only sessions are truly independent observations". Yes, sessions are independent observations. One might further ask to what extent are encounters or even individual rounds mutually independent? First, we believe that encounters are independent to a large degree because volunteers repeatedly (i.e., from an encounter to the next) exhibit very similar behaviours, e.g., (i) they start each

encounter with random action choice without realising that it would be better to systematically start with cooperation and then adjust according to what the opponent does, and (ii) the decoy's effectiveness is restored at the beginning of every new encounter despite wearing off through the course of the previous encounter. Nonetheless, encounters are not entirely independent because the slowly increasing fraction of tit-for-tat actions throughout sessions of the experiment signifies that some learning takes place (Fig. 4). As for individual rounds, we believe that they are very much interdependent because of the prevalence of tit-for-tat play and the fact that noise in tit-for-tat is biased.

Figure: Does noisy tit-for-tat play generate observable quantities that violate the assumptions of standard statistical tests? If this were the case, computer simulated recreations of the experiment, which incorporate noisy tit-for-tat (see SM, Computer simulations), should produce an empirical distribution function of the test statistic---in this case we look at Mann-Whitney U ---different from the test statistic assumed by theory and used to calculate p-values. We see that with the increasing number of simulations, the empirical distribution function approximates the theoretical one with high accuracy. Therefore, p-values calculated using the theoretical distribution function are the true probabilities of obtaining outcomes that are at least as extreme as the actual outcome of our experiment.

The reviewer concludes that: “the p-values reported[...] rest on an inflated number of independent observations.” Here, we urge the reviewer to reconsider how we define action frequencies in Fig. 1 and Fig. 3. We obtained these action frequencies by counting, for each volunteer, the number of cooperative, defecting, or rewarding actions taken, and then dividing these counts by the total number of rounds played. This definition means that “observations” are statements of the following type: volunteer A chose cooperation on average $xx\%$ of the time; volunteer B chose cooperation on average $yy\%$ of the time. *The question, therefore, is whether the observation that volunteer A chose cooperation $xx\%$ of the time is independent from the observation that volunteer B chose cooperation $yy\%$ of the time. Apart from meeting each other once throughout a session of the experiment, volunteers A and B are two separate individuals who were free to choose whichever action they wanted and, while doing so, they had no means of communicating their choices to one another.* In this sense, the above-mentioned observations are independent, making them suitable for statistical tests undertaken in the context of both Fig. 1 and Fig. 3.

To summarise our response in more mathematical terms, the reviewer's concern seems to

suggest that we are possibly conflating a set of independent, identically distributed (IID) random variables $X_i, i=1, \dots, n$ ---for which all the usual central limit theorems, laws of large numbers, and by extension statistical tests hold---with a set of serially dependent random variables $Y_t, t=1, \dots, T$ ---for which some central limit theorems and laws of large numbers exist, but much care is needed in applying them to statistical testing. A typical, but not the only, example of the latter category of random variables are time series. *Round-to-round data definitely belong to this latter category, hence the analysis in Fig. 2 uses time-series methodology. However, the overall cooperativeness of one volunteer is obtained rather independently from the overall cooperativeness of any other individual volunteer, suggesting that data in Fig. 1 and Fig. 3 are of the former category. This in turn justifies the use of performed statistical tests.*

We hope that these explanations clearly reflect our reasoning behind the conducted statistical analyses, that this reasoning is in agreement with the reviewer's viewpoint, and that the whole issue is hereby resolved to everyone's satisfaction. If the reviewer still feels that we should apply certain corrections to the p-values in Fig. 1 and Fig. 3, we would be glad to hear their suggestions.

* * *

Reviewer #2:

Comment: *First, let me apologize for the lateness of my review. I've reviewed the authors comments and gone over the changes to the manuscript, and I'm satisfied in their responses. They now do a much better job in fitting into the decoy literature, and I appropriate learning more about the focus upon dilemma strength.*

Reply: Thank you very much for your positive assessment. It means a lot.

Comment: *I'm still not sure that this focus provides for a completely natural analog to the single player games that decoys are usually tested in. R is certainly an inferior option, but the focus on the outcome, not the initial choice set, makes comparisons more difficult. In fact, I wonder if in this use, your R is actually similar to a symmetrically dominated decoy, or RF decoy from Wedell and Pettibone (1996), in that its fully dominated by both options. There's no dimension that you can reduce the comparison between R and D to in which a choice of R is better, and its the same with C. In the AD situation, the competitor is still superior to the decoy on a single dimension. I'll leave that determination up to the authors.*

Reply: In *single-shot games*, the reviewer's comment would be spot on: action D indeed dominates over action C , which in turn dominates over action R . This is seen from the bilateral payoff matrix in which payoffs associated with D (C) are higher than the corresponding payoffs associated with C (R). However, as we mentioned in a response to reviewer #1 above, *game repetitions* complicate matters. Because many elements of what would constitute an appropriate answer here can already be found in the said response to reviewer #1, we provide a summary.

Game repetitions have the potential to change the nature of the social dilemma by giving cooperative actions (C and R) a clear advantage over defection D , but only if the population contains enough cooperators to begin with. The critical fraction of cooperators needed is given by $x_c^* = q DS / (1 - q)$, where DS is dilemma strength and q is the game termination probability. In any case, the more cooperators there are, the more successful they get (in terms of their payoffs) relative to defectors. Action C , and even action R , thus become better than D as we get closer to the limit of a fully cooperative population.

In designing the present experiment, we were primarily inspired by the literature on asymmetrically dominated decoys, which is what we tried to recreate here, but in a context of interest within evolutionary game theory. In doing so, we operated under a number of practical constraints in order to keep this study comparable to the previous ones with slightly different focus (e.g., the role of reward here vs. the role of punishment in Ref. 29). Overall, we are very happy that the reviewer recognised the value of our efforts as reflected in his original comment on the novelty of the present approach.

Comment: *Overall, I'm happy with the improvement in the attempt to reconcile the two sets of literature. I support publication as is, after consideration of my comments.*

Reply: Thank you once again. We are equally grateful for your constructive suggestions and a positive overall evaluation. The former improved the manuscript, while the latter motivates us to keep up the good work.

REVIEWERS' COMMENTS:

Reviewer #1 (Remarks to the Author):

I think this revision has further improved the manuscript, which I now think is close to publishability. I accept the rebuttal of my statistical worries and appreciate the detailed answer to this issue, and all the others I raised as well. However, despite being in favor of publication, I have to admit that I am still not 100% happy with the paper.

On the new text on page 4, you write "... defection D dominates over cooperation C, which in turn dominates over reward R." I don't understand the phrase "in turn", which in my understanding has no theoretical meaning here. Both D and C dominate R. There is no iterated dominance here. This should be a very easy fix.

And then there is the issue of dilemma strength (I am sorry). Despite the rewriting, which has improved matters, I still don't understand the arguments, definitely not without consulting the SM; the expression on p.5, 9th line, is just meaningless if you don't know what DS is. When I read the SM, I understand what DS means (which, btw, is much older than ref 25 suggests, see, eg., Rapoport, J Conflict Res 1967). I agree with the authors that a full-fledged discussion of this in the main text is not useful. A compromise could be to have an abbreviated version (the formal definition, basically, eqns S2a,b) as part of the methods section. In addition, it could help if, on, p.5, third line, a verbal definition would be given, for instance, where you say "... whereby the stronger the dilemma", you could add a short verbal explanation of the form "that is, ..." where you say something along the line of what you say in the first couple of line on p.13 of the SM.

Reviewer #1:

Comment: *I think this revision has further improved the manuscript, which I now think is close to publishability. I accept the rebuttal of my statistical worries and appreciate the detailed answer to this issue, and all the others I raised as well. However, despite being in favor of publication, I have to admit that I am still not 100% happy with the paper.*

Reply: We are happy with the reviewer's positive evaluation, and wish to thank them for their effort in improving this manuscript.

Comment: *On the new text on page 4, you write "... defection D dominates over cooperation C , which in turn dominates over reward R ." I don't understand the phrase "in turn", which in my understanding has no theoretical meaning here. Both D and C dominate R . There is no iterated dominance here. This should be a very easy fix.*

Reply: We rewrote the part indicated by the reviewer: "[...] defection D dominates over cooperation C , and both D and C dominate over reward R ."

Comment: *And then there is the issue of dilemma strength (I am sorry). Despite the rewriting, which has improved matters, I still don't understand the arguments, definitely not without consulting the SM; the expression on p.5, 9th line, is just meaningless if you don't know what DS is. When I read the SM, I understand what DS means (which, btw, is much older than ref 25 suggests, see, eg., Rapoport, J Conflict Res 1967). I agree with the authors that a full-fledged discussion of this in the main text is not useful. A compromise could be to have an abbreviated version (the formal definition, basically, eqns S2a,b) as part of the methods section. In addition, it could help if, on, p.5, third line, a verbal definition would be given, for instance, where you say "... whereby the stronger the dilemma", you could add a short verbal explanation of the form "that is, ..." where you say something along the line of what you say in the first couple of line on p.13 of the SM.*

Reply: In accordance with the reviewer's comment, we added on page 5: "For example, by taking a difference between payoffs obtained for successful defection and mutual cooperation, and then normalising by the difference in payoffs obtained for mutual cooperation and mutual defection, we quantify how 'lucrative' defection is relative to cooperation."

This is a verbal version of the definition of one of the dilemma strength parameters as described in the Supplementary Methods, Dilemma Strength. With this definition, the readers should have easier time understanding the following sentence: "Accordingly, the stronger the dilemma, the easier it gets for defectors to exploit cooperators[...]" even without consulting the Supplementary Methods.

We emphasise that we had considered formally defining the dilemma strength parameters in the main text even before the reviewer raised this issue, but decided that doing so would be inadvisable. Namely, defining the two parameters would require introducing four mathematical symbols (R , S , T , and P), all of which would be used only once or, at most, twice. Furthermore, reward payoff R could very easily be confused with reward action R . We thus firmly believe that, instead of offering more information in the main text, the readers who want to learn more about dilemma strength should make the effort to check the Supplementary Methods, Dilemma Strength.